# Construction of Magnetic Composite Bacterial Carrier and Application in 17*β*-Estradiol Degradation

**DOI:** 10.3390/molecules27185807

**Published:** 2022-09-07

**Authors:** Sicheng Wu, Peng Hao, Zongshuo Lv, Xiqing Zhang, Lixia Wang, Wangdui Basang, Yanbin Zhu, Yunhang Gao

**Affiliations:** 1College of Animal Science and Technology, Jilin Agricultural University, Changchun 130118, China; 2Northeast Institute of Geography and Agroecology, Chinese Academy of Sciences, Changchun 130102, China; 3Institute of Animal Husbandry and Veterinary Medicine, Tibet Academy of Agricultural and Animal Husbandry Science, Lhasa 850009, China

**Keywords:** biodegradation, 17*β*-estradiol, immobilization

## Abstract

Estrogen contamination is widespread and microbial degradation is a promising removal method; however, unfavorable environments can hinder microbial function. In this study, a natural estrogen 17*β*-estradiol (E2) was introduced as a degradation target, and a new combination of bacterial carrier was investigated. We found the best combination of polyvinyl alcohol (PVA) and sodium alginate (SA) was 4% total concentration, PVA:SA = 5:5, with nano-Fe_3_O_4_ at 2%, and maltose and glycine added to promote degradation, for which the optimal concentrations were 5 g·L^−1^ and 10 g·L^−1^, respectively. Based on the above exploration, the bacterial carrier was made, and the degradation efficiency of the immobilized bacteria reached 92.3% in 5 days. The immobilized bacteria were reused for three cycles, and the degradation efficiency of each round could exceed 94%. Immobilization showed advantages at pH 5, pH 11, 10 °C, 40 °C, and 40 g·L^−1^ NaCl, and the degradation efficiency of the immobilized bacteria was higher than 90%. In the wastewater, the immobilized bacteria could degrade E2 to about 1 mg·L^−1^ on the 5th day. This study constructed a bacterial immobilization carrier using a new combination, explored the application potential of the carrier, and provided a new choice of bacterial immobilization carrier.

## 1. Introduction

Estrogen is a known endocrine disruptor, divided into natural (estrone (E1), 17*β*-estradiol (E2), and estriol (E3)) and synthetic (17*α*-ethinylestradiol (EE2)). Natural estrogen is mainly excreted by humans and animals [1], and due to the centralized breeding of livestock, animal excreta has become the main source of estrogen in the environment [2]. The annual emissions of estrogen from farms can reach 33 and 49 tons [3], respectively, and annual estrogen emissions from livestock can reach 83 tons in the US and EU [2]. Estrogen pollution is widespread due to large emissions, and different concentrations of estrogen can be detected in surface water, lagoon ponds, and grazing land water [4,5]. Estrogen mainly affects the reproductive system of organisms, such as the bisexual fish being found in estrogen-contaminated environments [6], but also can affect the development of organisms by causing endocrine disorders [7], and has been identified as a Class I carcinogen (http://monographs.iarc.fr/ENG/Classification/latest_classif.php (accessed on 9 August 2022)). Wastewater treatment plants (WWTP) are gathering sites for contaminated water; however, estrogens are still detected in the effluent [8,9]. This presence may be due to the focus of the WWTP on the removal of COD, total phosphorus, etc.; whereas the WWTP does not have a specific process for removing estrogen [10]. Nowadays, microbial removal is in focus as one of the economical and efficient ways of estrogen removal, and the complete degradation of estrogen has also been reported [1]. 17*β*-estradiol (E2), one of the natural estrogens, is notable for its high content and strong activity. To solve the E2 pollution problem, studies have isolated degrading bacteria, such as *Bacillus* sp., *Sphingomonas* sp., *Rhodococcu* sp., *Novosphingobium* sp., etc. [11,12,13]. In addition to bacteria, fungi can produce E2 degrading enzymes, e.g., ligninolytic enzymes produced by white rot fungi [14], and laccase from the white-rot fungus *Trametes hirsute* [15]. However, microorganisms will be affected by the unfavorable environment (unsuitable temperature, pH, lack of influencing substances, competition from other microorganisms) [16,17]. To make the microorganism maintain stable degradation activity, immobilization technology was proposed; immobilization can reduce external influences and prevent microbial loss by immobilizing microorganisms in a specific space [18]. Various materials are emerging as skeletons for tailoring microbial immobilization, such as silkworm excrement immobilization *Arthrobacter globiformis* D47 for diuron removal [19], *Trametes versicolor* immobilized on wood chips to remove pesticides from wastewater [20], *Bacillus velezensis* strain immobilization on polyvinyl alcohol (PVA)-sodium alginate (SA)-nano-ZnO to remove organic matter in slaughter wastewater [21]. Studies on E2 removal by immobilized microorganisms found SA immobilized *Novosphingobium* sp. ARI-1 degrades E1, E2, and E3 [22]; and cellulose triacetate immobilized *Sphingomonas* sp. AHC-F and *Sphingobium* sp. AX-B degrades E2 [23]. PVA and SA are commonly used as microbial carriers, as they have the advantages of economical and high mechanical strength [24]; in addition, they are also used in the manufacture of adsorption membranes [25] and wound dressing [26]. Research on PVA-SA immobilized microorganisms includes: the removal of high concentrations of PAHs from soil washing solutions containing TX-100 by PVA-SA immobilized degrading bacteria [27], PVA-SA immobilized *Brucella* G5 to improve plant growth and soil fertility under drought and saline-alkali soil conditions [28], and treatment of nitrogen-rich wastewater with PVA-SA immobilized anaerobic ammonia-oxidizing bacteria [29]. PVA-SA can also be combined with other substances to form a composite carrier, such as PVA-SA + biochar immobilized microorganisms enhanced the removal of nitrate, manganese, and phenol [30]; degradation of polycyclic aromatic hydrocarbons in soil washing wastewater by PVA-SA-nano-Al_2_O_3_ immobilized bacteria [31]; and PVA-SA + Fe_3_O_4_ immobilized bacteria to remove atrazine [32]. Although as far as we know, there are no studies on E2 degradation by PVA-SA immobilized microorganisms.

Nano-Fe_3_O_4_ has the advantages of low cost, small size, and good biocompatibility [33], and adding nano-Fe_3_O_4_ can make the carrier carry the magnetism that enables it to respond to external magnetic fields, thus facilitating the recovery of immobilized bacteria [34]. Lack of nutrients can limit microbial function [35]; studies found the application of PVA-SA with a carbon source made of polycaprolactone and peanut shell + PVA-SA improved microbial denitrification efficiency [35], and a porous solid carbon source prepared using lychee extract + PVA-SA enhanced microbial denitrification [36]. However, in the above study, the PVA-SA + carbon source constitutes an exogenous carbon source to nourish microorganisms, and is not used to immobilize microorganisms. Here, a new combination (PVA-SA + carbon, nitrogen sources, and nano-Fe_3_O_4_) was attempted to construct a bacterial carrier, using E2 as a degradation target, and ng previously isolated *Lysinibacillus* sp. GG242 (MZ027481.1) from cattle farm wastewater (solid-liquid separated and unfermented fresh wastewater; obtained from Guangze ecological pasture, Changchun, China) as a degrading bacterium. In this study, we first constructed the immobilized carrier, and then explored: (1) a suitable total concentration (PVA + SA) and the ratio of PVA and SA (P:S); (2) the appropriate nano-Fe_3_O_4_ concentration; (3) and suitable carbon and nitrogen sources, and their respective concentrations. Second, we further investigated the application potential of the carrier, and explored: (1) the E2 degradation ability of the immobilized bacteria; (2) the reusability of the immobilized bacteria and observed the changes in microspheres at each degradation cycle; (3) the E2 degradation ability of the immobilized bacteria in different environments; (4) and the ability of immobilized bacteria to degrade E2 in wastewater.

## 2. Results and Discussion

### 2.1. PVA + SA Total Concentration and P:S

PVA has a biological affinity, but the formed microspheres have high viscosity and poor mass transfer [37]; adding SA can improve the internal structure and mass transfer [38], and increasing SA also enhances the strength of the microspheres. However, the suitable combination of PVA + SA and P:S was not clear. Therefore, we conducted this research to determine the suitable combination for the bacterial carrier. The properties and shape of the microspheres are shown in Table 1 and Appendix A. At 2% total concentration and P:S = 9:1, the spheroidization and strength of the microspheres were the worst, and breakage numbers were the highest.

As the proportion of SA increased at the same total concentration, or the total concentration increased; the sphericity and strength gradually improved, and the breakage number gradually decreased. This means that the total concentration and proportion of SA are positively related to the properties of the microspheres. However, when the total concentration and the proportion of SA exceed a certain value, the microspheres will trail. At 4% total concentration, P:S = 5:5, the microspheres began to tail. At 6% and 8% total concentration, P:S = 7:3, the microspheres began to tail. At 10% total concentration, P:S = 9:1, the microspheres began to tail. These results show that the appropriate concentration and P:S play an important role in the shape of the microspheres. Similar results were found in other studies; increasing the proportion and concentration of SA can cause the microspheres to tail [39]. The microsphere tailing is due to the increased density, and high density can hinder the contact between the immobilized bacteria and substrate, thus affecting the degradation efficiency [40]; and because the sphere has the smallest surface area and highest space utilization, trailing also affects the space utilization of microspheres. Excluding the trailing scheme, the remaining schemes are as follows: 2% total concentration, P:S is 7:3–1:9; 4% total concentration, P:S is 9:1–5:5; 6% and 8% total concentration, P:S is 9:1. High or low concentrations of SA and PVA can affect spheroidization, stability, strength, etc. [39], so a high or low percentage of PVA or SA is not a good choice, and at 4% total concentration, P:S = 5:5 has the strongest strength among the remaining schemes. Therefore, the best scheme is 4% total concentration; P:S = 5:5.

### 2.2. Appropriate Nano-Fe_3_O_4_, Carbon, and Nitrogen Sources

The electrostatic interaction between the nano-Fe_3_O_4_ facilitates the immobilization of bacteria in the carrier [41] and the adsorption of contaminants [42]. The adsorption results of microspheres at different nano-Fe_3_O_4_ concentrations are shown in Table 2. At 1%, the number of adsorbed microspheres was 35, and as the nano-Fe_3_O_4_ increased, the microspheres were completely adsorbed. Because a high concentration may hinder the colonization of bacteria, we selected 2% to form the microspheres.

To select a suitable carbon and nitrogen source, the effect on degradation was evaluated. In different carbon sources, glucose, lactose, and maltose promoted degradation. E2 could be degraded to about 5 mg·L^−1^ on the 1st day and the degradation efficiency exceeded 84% (*p* < 0.01) (Figure 1a,b). As degradation proceeded, the highest degradation efficiency (95.17%) could be achieved by adding maltose on the 5th day and E2 degraded to 1.4 mg·L^−1^. Elevated degradation efficiency may be related to high biomass, glucose, lactose, and maltose promoting the growth of bacteria (Figure 1c); conversely, sucrose and starch inhibited bacterial growth, and this may be the reason for the decreased degradation efficiency. Similar results were found in other studies; degradation of bisphenol A was inhibited by adding sucrose [43], which shows that sucrose and starch may not be suitable substrates.

All nitrogen sources promoted E2 degradation; in the presence of glycine, the degradation efficiency was higher than the others. The E2 was degraded to 0.9 mg·L^−1^ with 96.8% degradation efficiency on the 5th day (Figure 1d,e; *p* < 0.01), and the biomass was higher than in the control group (Figure 1f). The highest degradation efficiency was obtained with the addition of maltose or glycine, but the biomass was not the highest. This may be due to the high degradative enzyme activity, and studies show that the addition of growth substrate will induce the expression of the enzyme [44] This phenomenon may be explained by the co-metabolism degradation of E2. Enzymes are more efficient than bacteria, and this mechanism may be an important way to remove trace levels of E2 [1]. After selecting the best carbon and nitrogen source, the appropriate concentration was explored. When the maltose was 5 g·L^−1^, bacteria showed the strongest degradation ability; the remaining E2 concentration was 1.6 mg·L^−1^ with 96.6% degradation efficiency on the 5th day (Figure 2a,b; *p* < 0.01). As the maltose concentration increased, the biomass increased (Figure 2c); however, the highest biomass did not achieve the highest degradation efficiency. This may be due to the high maltose concentration creating a comfortable environment. For glycine, at 10 g·L^−1^, the lowest E2 concentration (1.7 mg·L^−1^), and the highest degradation efficiency (96.5%) and biomass were achieved (Figure 2d–f). When glycine exceeded 10 g·L^−1^, biomass and degradation efficiency decreased, which may be due to the increase in intermediates as the glycine increase, which affects the biomass and thus degradation. The different results for maltose and glycine may be due to the different mechanisms involved. Previous studies showed that lack of nutrients is the main reason for the incomplete removal of pollutants [45]; therefore, adding the appropriate energy source will promote degradation. Based on the above exploration, the composite carrier was constructed, and the preparation process of the composite carrier is shown in Figure 3. The chemical composition and internal structure are shown in Appendix A, and the main elements of the carrier were Fe, C, O, and Ca. The internal structure of the composite carrier was irregular with cavities. In the next step, the carrier will be used to immobilize bacteria for the degradation experiments.

### 2.3. Degradation Performance and Reusability of Immobilized Bacteria

The results of the degradation of E2 by immobilized bacteria and free bacteria are shown in Figure 4a,b. The degradation efficiency of the immobilized bacteria reached 92.3% on the 5th day, indicating that the bacteria still maintain degradation capacity after immobilization. The results are consistent with previous studies showing that the immobilization has little effect on bacteria and maintains the degradation performance [46]. The degradation efficiency of immobilized bacteria was higher than free bacteria on the 1st day (*p* < 0.01). This may be due to storage affecting the activity of free bacteria, and this result also indicates that immobilization can maintain the activity of bacteria. From the 3rd day, the degradation efficiency of free bacteria was higher than that of the immobilized bacteria (*p* < 0.05). This may be due to the free bacteria having larger contact with E2, allowing for more adequate degradation. Empty microspheres can adsorb about 1 mg·L^−1^ E2, which shows that the adsorption capacity of the carrier is weak. Previous studies on strong adsorption capacity carriers, such as biochar, demonstrated they can adsorb large amounts of pollutants; however, long-term use and complete removal still rely on biodegradation [47]. Good reusability is one of the advantages of immobilized bacteria [48], and also contributes to the green development of the ecological environment [49]. In 3 degradation cycles, the degradation efficiency of each round exceeded 94% (Figure 5a), and as the degradation cycle increased, the internal structure of the microspheres was not cracked (Figure 5b). The degradation efficiency of immobilized bacteria was significantly higher than that of the free bacteria (*p* < 0.01), and as the cycle increases, the degradation efficiency of immobilized and free bacteria gradually increases. This may be due to the increase in biomass resulting in higher degradation efficiency. In this experiment, storage greatly affected the activity of the free bacteria, and the degradation efficiency was still lower than in the immobilized bacteria on the 3rd day, which further demonstrates the advantages of immobilization. Other studies found embedded *Enterobacter* sp. BRC05 with gellan gum can be reused for 9 cycles [50], *Bacillus cereus* WL08 immobilized with bamboo charcoal-SA for 10 cycles [51], and nano-Fe_3_O_4_ immobilization *Pseudomonas* sp. W1 for 3 cycles [34]. The differences in reusability may be related to the biomass, various carriers, and degradation conditions. In summary, the immobilized carrier can maintain the bacterial degradation ability and show good reusability.

### 2.4. Protective Effect of Immobilized Bacteria and the Degradation Effect in Wastewater

Immobilization can protect the strains from unfavorable conditions [52]. In this section, we evaluate the protective effect of the carrier under unfavorable conditions. The results of E2 degradation are shown in Figure 6. The unfavorable environment inhibited the degradation ability of free bacteria; among the five conditions, only pH 11 had a small effect on free bacteria, and the degradation efficiency reached 86.9% on the 5th day, indicating that the bacteria were resistant to alkali. After immobilization, the degradation efficiency increased significantly with 96.5% degradation efficiency on the 5th day (*p* < 0.01). Under the remaining four conditions, the initial stage of degradation showed a consistent result. Both free and immobilized bacteria were in the adaptation period, and as degradation proceeded, the degradation efficiency of immobilized bacteria was significantly (*p* < 0.01) higher than that of the free bacteria on the 3rd day, and the degradation efficiency exceeded 90% on the 5th day. This indicates that the carrier can protect the bacteria well. The results are consistent with previous studies, in which the degradation efficiency of immobilized bacteria was higher than that of the free bacteria in unfavorable environments [53,54]. The higher degradation efficiency of immobilized bacteria may be due to the following reasons: first, the bacteria are embedded inside the carrier to increase the stability of the bacteria and avoid direct contact of the bacteria with harsh environments [55]; second, a stable environment also contributes to biofilm formation [55]; third, other nutrients are present in the carrier to promote the degradation and provide more energy; and fourth, immobilization. The ability to survive in unfavorable environments affects the prospects of microbial applications [34]. In this study, the carrier was used to immobilize the bacteria, and the carrier can protect bacteria well in unfavorable environments.

Wastewater is characterized by high COD, P, and N; and if it enters natural water, it leads to rapid depletion of dissolved oxygen and eutrophication of the water [56]. The degradation results and the changes in water quality indicators are shown in Figure 7a,b; the immobilized bacteria can maintain the degradation ability and can degrade E2 to 1mg·L^−1^ on the 5th day. For COD, COD was higher than the control group (*p* < 0.01), and showed a trend of increasing first and then decreasing. COD is used as one of the comprehensive indicators of the relative content of organic matter. The increase may be due to three reasons: first, the immobilized carrier is organic; second, the secretion of bacteria may contain organic components; and third, E2 became the intermediate product. The decrease in COD may be due to a shift in bacterial attention from E2 to other substances in the wastewater as E2 decreases. For total phosphorus (P), it was higher than the control group; but showed a decreasing trend, which may be due to P as a nutrient for bacteria. For ammonia nitrogen and nitrate nitrogen, in the control group, ammonia nitrogen gradually decreased, and nitrate nitrogen gradually increased, indicating that nitrification reactions may exist. After adding immobilized bacteria, both ammonia nitrogen and nitrate nitrogen increased. The increase in nitrate nitrogen may be due to the enhanced nitrification reaction; nitrification reaction is an oxidation reaction, and nano-Fe_3_O_4_ is oxidizing, so it will facilitate the reaction. The increase in ammonia nitrogen may be due to the decomposition of amino acids in the immobilized carrier, and ammonia nitrogen did not decrease in the control group, and this may be because nitrification is weaker than ammonification. In short, the carrier can maintain the E2 degradation ability of bacteria in wastewater. The above research reveals the potential of the carrier and lays a foundation for its practical application.

## 3. Materials and Methods

### 3.1. Reagents and Mediums

E2 (≥98%; 1g; CAS: 50-28-2) were purchased from Solarbio (Beijing, China), acetonitrile (high performance liquid chromatography (HPLC) grade; 99%; CAS:75-05-8) and methanol (HPLC grade; 99%; CAS: 67-56-1) from Thermo Fisher Scientific (Shanghai, China), nano-Fe_3_O_4_ (CAS: 1317-61-9) from Andi Metal Material (Heibei, China), and polyvinyl alcohol 1788 (PVA; CAS: 9002-89-5) and sodium alginate (SA; CAS: 9005-38-3) from Aladdin (Beijing, China). The Basal salt medium (BSM) was: 1.0 g·L^−^^1^ K_2_HPO_4_, 0.5 g·L^−1^ KH_2_PO_4_, 0.2 g·L^−1^ MgSO_4_, and 1.0 g·L^−1^ (NH_4_)_2_SO_4_. The Luria-Bertani medium (LB) was: 10.0 g·L^−1^ tryptone, 10.0 g·L^−1^ NaCl, 5.0 g·L^−1^ yeast extract. Unless otherwise specified, other reagents were analytical grade. 

### 3.2. Selection of PVA + SA Total Concentration and P:S

To explore the appropriate material concentration and P:S, different total concentrations (2–10%; *w*/*v*), and P:S (9: 1-1: 9; *w*/*w*) were set at the same concentration. The method of immobilizing bacteria is as described in [40]. Briefly, first, PVA was dissolved in water at 100 °C, and after complete dissolution, SA was added and stirred continuously until dissolved, and then cooled to room temperature and the bubbles removed. Subsequently, the material was dropped into the cross-linking solution (2% (*w*/*v*) CaCl_2_-saturated boric acid) and cross-linked for 24 h. The cross-linking solution was removed and the microspheres washed 3 times with sterile water, at 4 °C. The appropriate material was selected at a ratio according to the microspheres’ morphology, strength, and number of breakages in different environments (3 weeks at 200 rpm·min^−1^, pH 5, pH 11, 20 °C, and 40 °C).

### 3.3. Selection of Nano-Fe_3_O_4,_ Carbon, and Nitrogen Sources

Different concentrations (1–10%; *w*/*v*) of nano-Fe_3_O_4_ were added to form microspheres, and 40 microspheres were randomly selected for adsorption; and the concentration was selected based on the number adsorbed. To select a suitable carbon and nitrogen source, 1 g·L^−1^ carbon sources (sucrose, starch, glucose, lactose, maltose) and 1 g·L^−1^ nitrogen sources (yeast extract fermentation (YEF), tryptone, peptone, urea, glycine) were added to the BSM, respectively. The bacterial suspension was prepared as described in [57] with minor modifications. First, the strain was grown to log phase in LB, and the supernatant removed after centrifugation (7000 rpm·min^−1^, 5 min), washed 3 times with sterile water, and OD_600_ adjusted to 2.0, and the bacterial suspension was obtained. The degradation experiment was as described in [58] with minor modifications. The bacterial suspension was added to BSM to degrade E2 under the above conditions, and the best carbon and nitrogen source was selected according to the degradation results. After the carbon and nitrogen source selection was completed, the appropriate carbon and nitrogen source concentrations (0–20 g·L^−1^) were explored, and 50 mg·L^−1^ E2 was degraded under different concentrations of carbon and nitrogen sources, and the best concentration was chosen based on the results. Based on the above exploration, the carrier construction was completed and the carrier without bacteria was tested by scanning energy dispersive spectrometer (EDS) and electron microscope (SEM).

### 3.4. Evaluate E2 Degradation Performance and Reusability of Immobilized Bacteria

To evaluate the ability of immobilized bacteria to degrade E2, the bacterial suspension was prepared as described in Section 3.3, and mixed with glycine, maltose, and nano-Fe_3_O_4_ with the embedding material. After the bubbles were removed, the suspension was dropped into the cross-linking solution, and cross-linked for 24 h. The remaining suspension was stored at room temperature during cross-linking. Next, the cross-linking solution was removed and the microspheres washed 3 times with sterile water. Finally, the microspheres and the suspension were added to the LB for activation. After activation, the microspheres were washed 3 times with sterile water, and the activated bacteria were re-prepared into a bacterial suspension. Experimental groups: group 1: free bacteria + E2; group 2: immobilized bacteria (immobilized an equal volume of bacteria as free bacteria) + E2; group 3: microspheres without bacteria + E2; control group: E2. The reusability experiment was as described in [49] with modifications; 3 cycles of E2 degradation were set, 3 days each, and the microspheres and bacteria were filtered out after each round and washed 3 times with sterile water and then re-added to a new medium to start a new round of degradation. Experimental groups: group 1: free bacteria + E2; group 2: immobilized bacteria (immobilized an equal volume of bacteria as free bacteria) + E2; control group: E2. Sampling was conducted on the 1st and 3rd day of each cycle to detect the remaining E2, and the microspheres were taken out at the end of each cycle to observe the internal morphology of the microspheres via SEM.

### 3.5. Evaluate the Protective Properties of Immobilized Bacteria and the Application in Wastewater

The protective effect of immobilization was evaluated by the degradation efficiency of E2 in adverse environments. The bacterial suspension and immobilized bacteria were prepared as described in Section 3.3 and Section 3.4, and the experimental conditions were pH 5, pH 11, 20 °C, 40 °C, and 40 g·L^−1^ NaCl (conditions selected based on previous experimental results); only one condition was different per group, and the rest were the same. Each condition was divided into 3 groups; group 1: free bacteria + E2; group 2: immobilized bacteria (immobilized an equal volume of bacteria as free bacteria) + E2; control group: E2. Next, to evaluate the application performance of the immobilized bacteria in wastewater, E2 degradation experiments were performed in wastewater (solid-liquid separated and unfermented fresh wastewater; obtained from Guangze ecological pasture, Changchun, China). First, the wastewater was left to stand, the supernatant diluted 10 times and placed in the autoclave (conditions selected based on pre-experiment). Next, the immobilized bacteria were added to 100 mL wastewater containing E2 for degradation, and the E2, COD, NH_3_-N, NO_3−_-N, and total P were detected after each sample.

### 3.6. Devices and Analytical Methods

Detection of E2 was performed by HPLC, with the following HPLC settings: the instrument was a Shimadzu LC-2030 Plus (Kyoto, Japan); chromatographic column an Agilent ZORBAX SB-C18 (250 mm × 4.6 mm, 5 μm, Santa Clara, CA, USA); mobile phase acetonitrile: water = 55:45 (*v*/*v*) with a flow rate of 1.0 mL·min^−1^; column oven temperature 30 °C, injection volume 25 μL; and the detection wavelength 280 nm. The water quality detector (TR 6900), digestion instrument (TDR-25), and water quality testing reagents were purchased from Shenzhen Tongao (Shenzhen, China). Data were statistically analyzed by one-way ANOVA using SPSS software (version 25.0); and the graphs prepared using GraphPad Prism (version 7.0). Unless otherwise indicated, the above experiments were performed in 100 mL BSM containing 30 mg·L^−1^ E2 at 30 °C, 135 rpm·min^−1^ for 5 days; in triplicate, the bacterial suspension was added at 1% (*v*/*v*), and no bacteria as the control, 5-day degradation period, and sampling was conducted on the 1st, 3rd, and 5th day to detect the remaining E2.

## 4. Conclusions

In this study, a new combination (PVA-SA + nano-Fe_3_O_4_, and maltose, glycine sources) was used to build a bacterial carrier, and then the carrier was used to immobilize bacteria. The immobilized bacteria, which can maintain a stable degradation ability, were reused 3 times, and the degradation ability did not decrease. In unfavorable conditions (pH 5, pH 11, 20 °C, 40 °C, and 40 g·L^−1^ NaCl), the carrier shows strong protective properties. The degradation efficiency of the immobilized bacteria was significantly higher than that of the free bacteria, and in the wastewater environment, the immobilized bacteria also showed stable degradation ability. This study provides a new choice of bacterial immobilization carrier and lays the foundation for the application of the carrier in the actual environment. Future research should focus on combining the carrier with wastewater treatment equipment with a view to contributing to the research in a real environment.

## Figures and Tables

**Figure 1 molecules-27-05807-f001:**
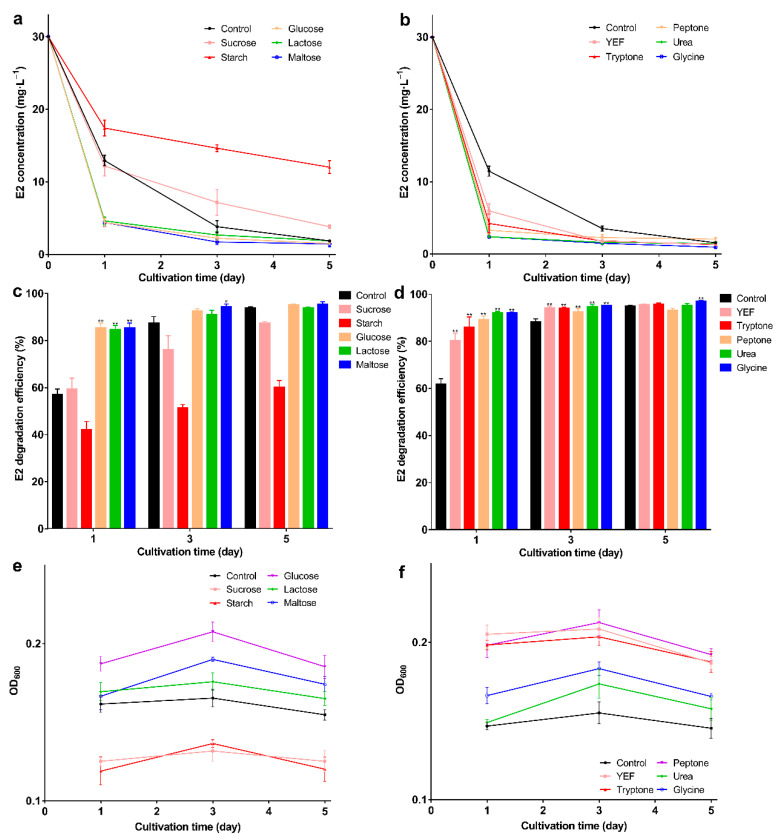
E2 residual concentration, degradation efficiency, and OD_600_ under different carbon and nitrogen sources; (**a**) E2 residual concentration; (**b**) E2 degradation efficiency; (**c**) OD_600_ under different carbon sources; (**d**) E2 residual concentration; (**e**) E2 degradation efficiency; (**f**) OD_600_ under different nitrogen sources. Data points are the average and error bars represent the standard errors of the three experiments; * (0.01 < *p* < 0.05); ** (*p* < 0.01).

**Figure 2 molecules-27-05807-f002:**
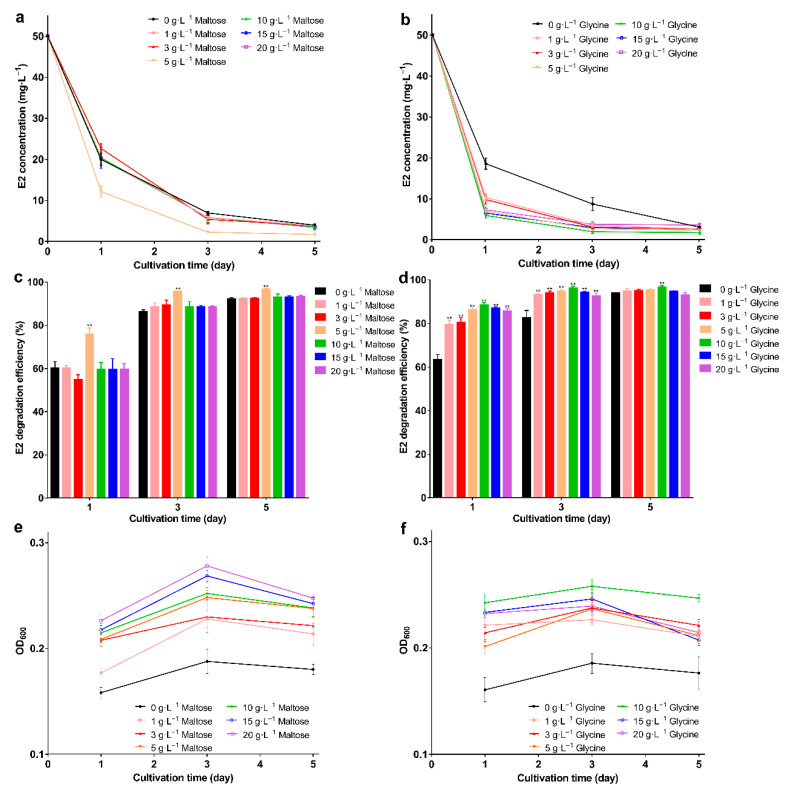
E2 residual concentration, degradation efficiency, and OD_600_ under different concentrations of maltose and glycine; (**a**) E2 residual concentration; (**b**) E2 degradation efficiency; (**c**) OD_600_ under different concentrations of maltose; (**d**) E2 residual concentration; (**e**) E2 degradation efficiency; (**f**) OD_600_ under different concentrations of glycine. Data points are the average and error bars represent the standard errors of the three experiments; ** (*p* < 0.01).

**Figure 3 molecules-27-05807-f003:**
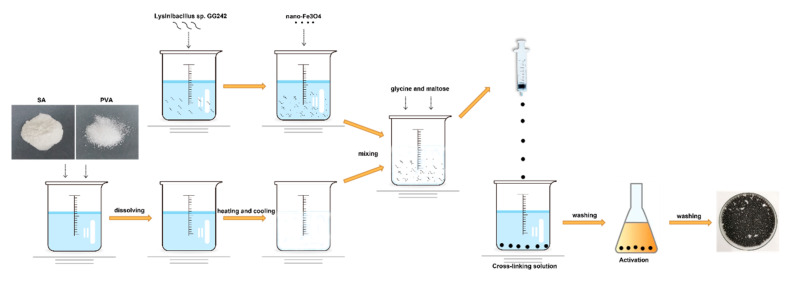
Preparation process of the composite carrier.

**Figure 4 molecules-27-05807-f004:**
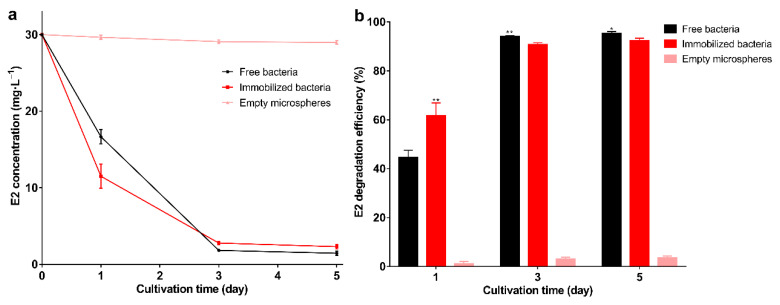
(**a**) E2 residual concentration; (**b**) E2 degradation efficiency under free bacteria, immobilized bacteria, and empty microspheres. Data points are the average and error bars represent the standard errors of the three experiments; * (0.01 < *p* < 0.05); ** (*p* < 0.01).

**Figure 5 molecules-27-05807-f005:**
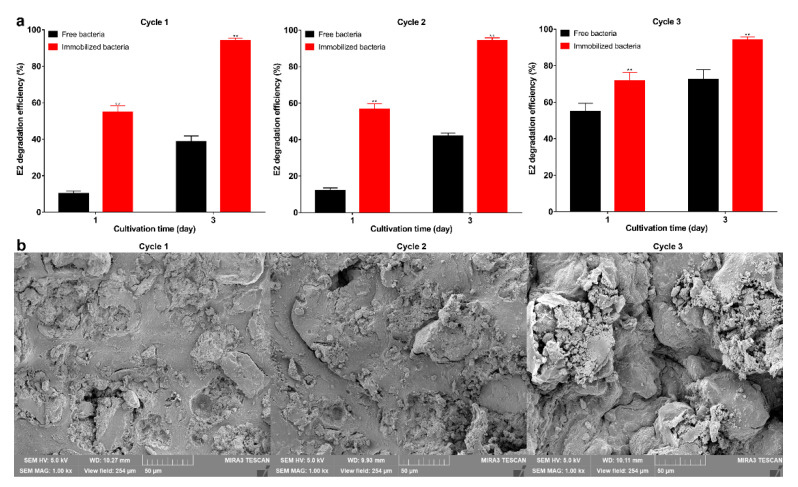
(**a**) Degradation efficiency of immobilized bacteria and free bacteria per cycle; (**b**) SEM image of the immobilized bacteria internal structure per cycle (1000×); scale bar is 50 μm. Data points are the average and error bars represent the standard errors of the three experiments; ** (*p* < 0.01).

**Figure 6 molecules-27-05807-f006:**
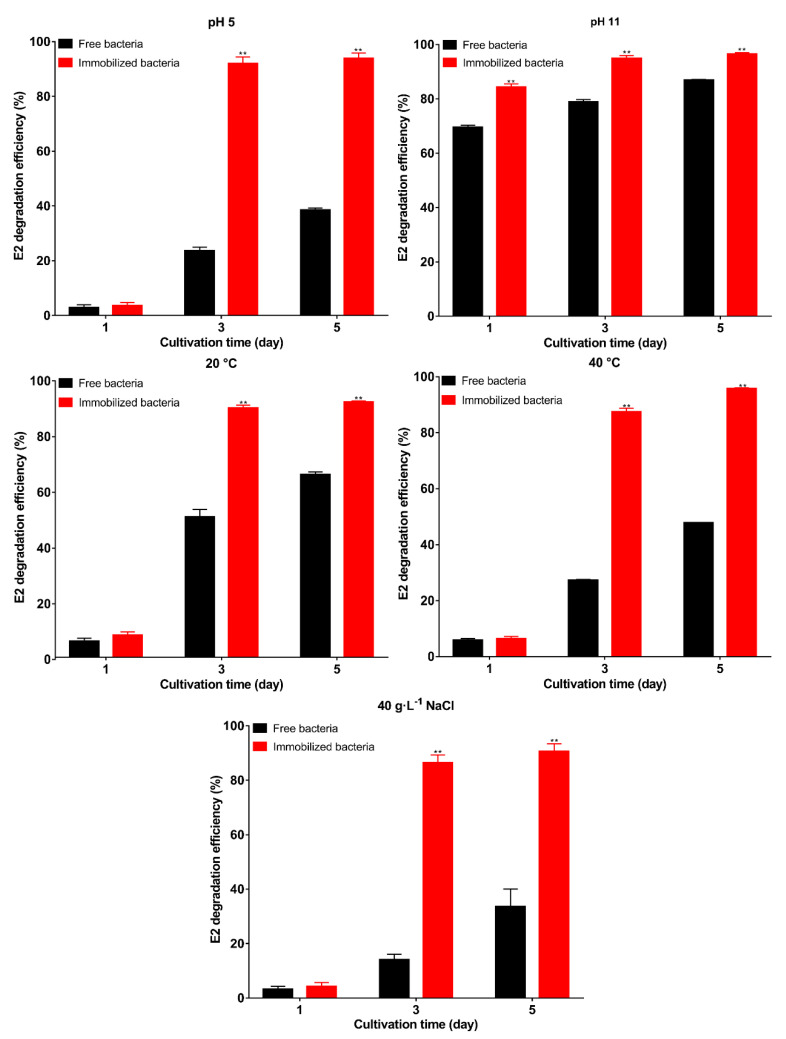
The E2 degradation efficiency of immobilized bacteria and free bacteria under pH 5, pH 11, 20 °C, 40 °C, and 40 g·L^−1^ NaCl. Data points are the average and error bars represent the standard errors of the three experiments; ** (*p* < 0.01).

**Figure 7 molecules-27-05807-f007:**
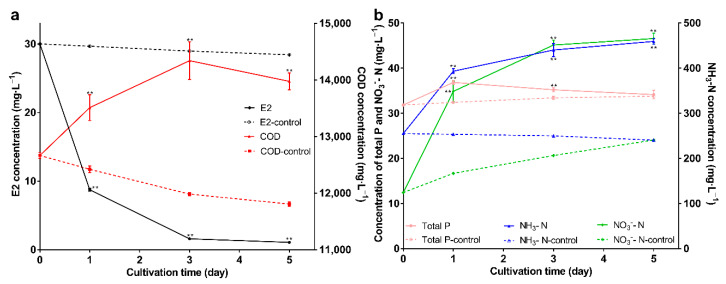
(**a**) The changes of E2 and COD; (**b**) The changes of NH_3_-N, NO_3_^−^-N, and total P. Data points are the average and error bars represent the standard errors of the three experiments; ** (*p* < 0.01).

**Table 1 molecules-27-05807-t001:** Properties of microspheres at different ratios.

P + S	P:S	Sphericity	Strength	Tailing	Number of Broken				
					200 rpm min^−1^	pH 5	pH 11	20 °C	40 °C
2%	9:1	−	80 g	−	40	40	40	40	40
	7:3	+	175 g	−	2	3	2	3	0
	5:5	+	480 g	−	0	0	0	0	0
	3:7	+	570 g	−	0	0	0	0	0
	1:9	+	720 g	−	0	0	0	0	0
4%	9:1	+	130 g	−	3	0	0	0	0
	7:3	+	440 g	−	0	1	0	0	0
	5:5	+	760 g	−	0	0	0	0	0
	3:7	−	1300 g	++	0	0	0	0	0
	1:9	−	1845 g	+++	0	0	0	0	0
6%	9:1	+	230 g	−	1	0	0	0	0
	7:3	+	650 g	+	0	0	0	0	0
	5:5	−	1050 g	++	0	0	0	0	0
	3:7	−	1450 g	++	0	0	0	0	0
	1:9	−	>2000 g	+++	0	0	0	0	0
8%	9:1	+	300 g	−	0	0	0	0	0
	7:3	−	810 g	++	0	0	0	0	0
	5:5	−	1450 g	++	0	0	0	0	0
	3:7	−	1800 g	+++	0	0	0	0	0
	1:9	−	>2000 g	+++	0	0	0	0	0
10%	9:1	+	350 g	+	0	1	0	0	0
	7:3	−	1000 g	++	0	0	0	0	0
	5:5	−	1750 g	+++	0	0	0	0	0
	3:7	−	>2000 g	+++	0	0	0	0	0
	1:9	−	>2000 g	+++	0	0	0	0	0

PVA: polyvinyl alcohol. SA: sodium alginate +: form microspheres/tails −: no form microspheres/tails.

**Table 2 molecules-27-05807-t002:** The number of microspheres adsorbed at different nano-Fe_3_O_4_ concentrations.

Nano-Fe_3_O_4_	1%	2%	3%	4%	5%	6%	7%	8%	9%	10%
Number of adsorbed	35	40	40	40	40	40	40	40	40	40

## Data Availability

The datasets generated and analyzed during the current study are available from the corresponding author on reasonable request.

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
