# Peer review of "Construction of Magnetic Composite Bacterial Carrier and Application in 17β-Estradiol Degradation"

_molecules, 2022, doi:10.3390/molecules27185807_

Round 1

Reviewer 1 Report

Please improve the state of the art analysis to show the progress beyond the state of the art clearly. The lack of proper justification creates the wrong impression that the authors are unaware of the recent developments. Please use relevant recent references by OTHER authors, recent meaning from 2021 and 2022.

The title needed significant modification.

Introduction: They should summarise research findings and novelty. The purpose of the study is not well expressed. I have a problem to find what is the unique and original research novelty. Would you please be more explicate about that?

Results and discussion: line 69: I could not find Table S1,

Line 136-14: its look like methods, Figure S1 also missing from the manuscript,

What is the use of micrograph in between figure 4 and figure 5, work on your figure presentation, You may publish some of those as Supplementary Materials. Method or experimental section should be cited by proper references. Now is unacceptable. Where the authors get this method and this condition?

Materials ans methods: reported methods are developed bu the authors or they modified from published methods, it is not clear since is there no any references in this section

Conclusion is too short and is not informative.

Future scope of this study can be added as well as social impact can also be discussed in this paper.

Author Response

Response to Reviewer 1 Comments

Point 1: Please improve the state of the art analysis to show the progress beyond the state of the art clearly. The lack of proper justification creates the wrong impression that the authors are unaware of the recent developments. Please use relevant recent references by OTHER authors, recent meaning from 2021 and 2022.

Response 1: Revisions have been made in the manuscript.

Point 2: The title needed significant modification.

Response 2: Revisions have been made in the manuscript.

Point 3: Introduction: They should summarise research findings and novelty. The purpose of the study is not well expressed. I have a problem to find what is the unique and original research novelty. Would you please be more explicate about that?

Response 3: Revisions have been made in the manuscript. We would like to try a new combination to construct bacterial immobilization carriers and this is the first time to use sodium alginate and polyvinyl alcohol to immobilize bacteria to degrade 17β-estradiol.

Point 4: Results and discussion: line 69: I could not find Table S1.

Response 4: We are sorry for this issue, we placed Table S1 in the supplementary data and re-uploaded, you can find it in Supplementary Data.

Point 5: Line 136-14: its look like methods, Figure S1 also missing from the manuscript,

Response 5: Revisions have been made in the manuscript. We placed Figure S1 in the supplementary data and re-uploaded, you can find it in Supplementary Data.

Point 6: What is the use of micrograph in between figure 4 and figure 5, work on your figure presentation, You may publish some of those as Supplementary Materials. Method or experimental section should be cited by proper references. Now is unacceptable. Where the authors get this method and this condition?

Response 6: Revisions have been made in the manuscript.

Point 7: Materials ans methods: reported methods are developed bu the authors or they modified from published methods, it is not clear since is there no any references in this section

Response 7: Revisions have been made in the manuscript.

Point 8: Conclusion is too short and is not informative.

Response 8: Revisions have been made in the manuscript.

Point 9: Future scope of this study can be added as well as social impact can also be discussed in this paper.

Response 9: We have supplemented in the manuscript.

Reviewer 2 Report

Review Report of molecules-1885239  

In the manuscript entitled Immobilization of Lysinibacillus sp. GG242 with a magnetic composite carrier enhances the degradation of 17β-estradiol investigated the immobilization of Lysinibacillus sp. GG242 with a magnetic composite carrier to enhance the degradation of 17β-estradiol. The study is backed up with high class experimental data and evidences, which are currently followed for similar types of work worldwide. In totality, the conceptualization, designing of experiments and the overall write up is good and quite clear.

However, it needs major corrections and there are some queries which the authors should kindly respond to make it good.

Major comments:

1. Errors in grammar and language editing. Authors are responsible for preparing their papers in correct English language. The manuscript requires substantial grammatical revisions in its present form. Both the technical and grammatical revisions should be made and the English should be polished.

Technical queries/suggestions:

1. Abstract: please use degradation efficiency instead of degradation rate. Check and revise throughout the manuscript.

2. Abstract: Is this composite bacterial carrier used for biodegradation before? Or novel? The abstract section should be written more precisely and explain novelty of this work.

3. Introduction Line 39: Are there any fungal strains used for estrogen degradation? Please add this information.

4. Introduction Line 50: ..isolated Lysinibacillus sp. GG242 (MZ027481.1) was used as a degrading bacterium. Please add more details for the isolated Lysinibacillus sp. GG242 (MZ027481.1). Where is it from? Has Lysinibacillus sp. GG242 described strain been deposited in a public strain collection? Please also mention the collection number in the manuscript.

5. The Introduction section needs to be reorganized. The authors should add the information in this section why the authors select Lysinibacillus sp. GG242 in the experiment? Why you select the immobilization of Lysinibacillus sp. GG242 with a magnetic composite carrier? In addition, the Introduction section should briefly place the study in a broad context and highlight why it is important. It should define the purpose of the work and its significance. The current state of the research field should be reviewed, and key publications cited. Finally, briefly mention the main aim of the work and highlight the principal conclusions. However, the novelty and significance of the manuscript were not highlighted in the Introduction section, please modify the introduction more clearly.

6. Results and discussion: Figure 5. Please add the magnification times SEM figures and described details in the figure captions.

7. Materials and Methods: 3.3. Selection of nano-Fe3O4, carbon, and nitrogen sources. Authors should include the citation to support the method. The same as 3.4 and 3.5.

8.  Conclusion: Authors can add and revise this section for the better understanding of the topic and its future research.

9. References: Please check the references carefully. Use italic for all the genus names.  

10.The English can be improved. Please have your manuscript checked by a technical editor.

Author Response

Response to Reviewer 2 Comments

Point 1: Errors in grammar and language editing. Authors are responsible for preparing their papers in correct English language. The manuscript requires substantial grammatical revisions in its present form. Both the technical and grammatical revisions should be made and the English should be polished.

Response 1: Revisions have been made in the manuscript.

Point 2: Abstract: please use degradation efficiency instead of degradation rate. Check and revise throughout the manuscript.

Response 2: Revisions have been made in the manuscript.

Point 3: Abstract: Is this composite bacterial carrier used for biodegradation before? Or novel? The abstract section should be written more precisely and explain novelty of this work.

Response 3: Revisions have been made in the manuscript.

Point 4:  Introduction Line 39: Are there any fungal strains used for estrogen degradation? Please add this information.

Response 4: We have supplemented in the manuscript.

Point 5: Introduction Line 50: ..isolated Lysinibacillus sp. GG242 (MZ027481.1) was used as a degrading bacterium. Please add more details for the isolated Lysinibacillus sp. GG242 (MZ027481.1). Where is it from? Has Lysinibacillus sp. GG242 described strain been deposited in a public strain collection? Please also mention the collection number in the manuscript.

Response 5: We have supplemented in the manuscript and the strain is not deposited in the public strain repository.

Point 6: The Introduction section needs to be reorganized. The authors should add the information in this section why the authors select Lysinibacillus sp. GG242 in the experiment? Why you select the immobilization of Lysinibacillus sp. GG242 with a magnetic composite carrier? In addition, the Introduction section should briefly place the study in a broad context and highlight why it is important. It should define the purpose of the work and its significance. The current state of the research field should be reviewed, and key publications cited. Finally, briefly mention the main aim of the work and highlight the principal conclusions. However, the novelty and significance of the manuscript were not highlighted in the Introduction section, please modify the introduction more clearly.

Response 6: Revisions have been made in the manuscript. This strain was isolated from previous experiments, and follow-up studies were carried out due to less research on its degradation of 17β-estradiol. The content of its identification and degradation properties is in another article and this article is also in the submission period.

Point 7: Results and discussion: Figure 5. Please add the magnification times SEM figures and described details in the figure captions.

Response 7: We have supplemented in the manuscript.

Point 8: Materials and Methods: 3.3. Selection of nano-Fe3O4, carbon, and nitrogen sources. Authors should include the citation to support the method. The same as 3.4 and 3.5.

Response 8: We have supplemented in the manuscript.

Point 9: Conclusion: Authors can add and revise this section for the better understanding of the topic and its future research.

Response 9: We have supplemented in the manuscript.

Point 10: References: Please check the references carefully. Use italic for all the genus names. 

Response 10: Revisions have been made in the manuscript.

Point 11: The English can be improved. Please have your manuscript checked by a technical editor.

Response 11: Revisions have been made in the manuscript.

Round 2

Reviewer 1 Report

All the questions asked by the me have been well addressed and the manuscript has been significantly improved by this major revision.there is some problems like references in the text, typo, editing issues etc.

Reviewer 2 Report

Review Report of molecules-1885239-V2  

Although authors have provided a response to the comments, it is suggested to submit high resolution figures. All the figures are not clear in the present form.

It is difficult to find which correction has been performed at which place in the manuscript, so it is suggested to revise response to reviewers file indicating the line nos. For the reply of each comment.

Add the main results of this study into the abstract.

The immobilization technique, using natural or synthetic materials to restrict the high density of degraders to a fixed space, provides bacteria a protected environment in which to survive, slows the microbial dispersal speed, exhibits improved removal efficiency of target compounds, and is reusable.Various materials are emerging as skeletons for tailoring microbial immobilization, such as silkworm excrement, sodium alginate, cereal straw, farmyard manure, press mud compost, fresh cow dung, and gypsum. It is suggested to add recent references about the immobilization technique successfully used for improving removal efficiency of target compounds to support your statement in the Introduction section. The following papers belong to this topic, I hope they are useful for you.

Liu J, et al. Enhanced diuron remediation by microorganism-immobilized silkworm excrement composites and their impact on soil microbial communities. J Hazard Mater. 2019 Aug 15;376:29-36.

Wahla AQ, et al. Immobilization of metribuzin degrading bacterial consortium MB3R on biochar enhances bioremediation of potato vegetated soil and restores bacterial community structure. J Hazard Mater. 2020 May 15;390:121493.

Bhatt P, et al. Indigenous bacterial consortium-mediated cypermethrin degradation in the presence of organic amendments and Zea mays plants. Environ Res. 2022 Sep;212:113137.   

Beltrán-Flores E, et al. Fungal bioremediation of agricultural wastewater in a long-term treatment: biomass stabilization by immobilization strategy. J Hazard Mater. 2022 Jul 21;439:129614.

Follow guide for authors strictly.
